



# Ground-based FTIR O₃ retrievals from the 3040 cm⁻¹ spectral range at Xianghe, China

Minqiang Zhou[1], Pucai Wang[2,3,4], Bavo Langerock[1], Corinne Vigouroux[1], Christian Hermans[1], Nicolas Kumps[1], Ting Wang[2], Yang Yang[2], Denghui Ji[2], Liang Ran[2], Jinqiang Zhang[2], Yuejian Xuan[2], Hongbin Chen[2,3,4], Françoise Posny[5], Valentin Duflot[5,6], Jean-Marc Metzger[6], and Martine De Mazière[1]

[1]Royal Belgian Institute for Space Aeronomy (BIRA-IASB), Brussels, Belgium
[2]Key Laboratory of Middle Atmosphere and Global Environment Observation, Institute of Atmospheric Physics, Chinese Academy of Sciences, Beijing, China
[3]University of Chinese Academy of Sciences, Beijing, China
[4]Xianghe Observatory of Whole Atmosphere, Institute of Atmospheric Physics, Chinese Academy of Sciences, Xianghe, China
[5]LACy, Laboratoire de l'Atmosphère et des Cyclones, UMR8105 (CNRS, Université de La Réunion, Météo-France), Saint-Denis, Réunion, France
[6]UMS 3365 – OSU Réunion, Université de La Réunion, Saint-Denis, Réunion, France

**Correspondence:** Minqiang Zhou (minqiang.zhou@aeronomie.be), Pucai Wang (pcwang@mail.iap.ac.cn)

**Abstract.** In this study, we present O₃ retrievals from ground-based Fourier-transform infrared (FTIR) solar absorption measurements between June 2018 and December 2019 at Xianghe, China (39.75 °N, 116.96 °E). The FTIR spectrometer at Xianghe is operated with indium gallium arsenide (InGaAs) and indium antimonide (InSb) detectors, recording the spectra between 1800 and 11000 cm⁻¹. As the harmonized FTIR O₃ retrieval strategy (Vigouroux et al., 2015) within the Network for the Detection

5  of Atmospheric Composition Change (NDACC) uses the 1000 cm⁻¹ spectral range, we designed an alternative O₃ retrieval strategy in the 3040 cm⁻¹ spectral range at Xianghe.

The retrieved O₃ profile is mainly sensitive to the vertical range between 5 and 40 km, and the degree of freedom for signal is 2.4±0.3 (1σ), indicating that there are two individual pieces of information in partial columns between the surface and 20 km and between 20 and 40 km. According to the optimal estimation method, the systematic and random uncertainties of the

10  FTIR O₃ total columns are about 13.6% and 1.4%, respectively. The random uncertainty is consistent with the observed daily standard deviation of the FTIR retrievals.

To validate the FTIR O₃ total and partial columns, we apply the same O₃ retrieval strategy at Maïdo, Reunion Island (21.08 °N, 55.38 °E). The FTIR O₃ (3040 cm⁻¹) measurements at Xianghe and Maïdo are then compared with the nearby ozonesondes at Beijing (39.81 °N, 116.47 °E) and at Gillot (20.89 °S, 55.53 °E), respectively, as well as with co-located

15  TROPOspheric Monitoring Instrument (TROPOMI) satellite measurements at both sites. In addition at Maïdo, we compare the FTIR O₃ (3040 cm⁻¹) retrievals with the standard NDACC FTIR O₃ measurements using the 1000 cm⁻¹ spectral range. It is found that the total columns retrieved from the FTIR O₃ 3040 cm⁻¹ measurements are underestimated by 5.5 - 9.0 %, which is mainly due to the systematic uncertainty in the partial column between 20 and 40 km (about -10.4%). The systematic uncertainty in the partial column between surface and 20 km is relatively small (within 2.4%). By comparison with other





measurements, it is found that the FTIR $O_3$ (3040 cm$^{-1}$) retrievals capture very well the seasonal and synoptic variations of the $O_3$ total and two partial columns. Therefore, the ongoing FTIR measurements at Xianghe can provide useful information on the $O_3$ variations and (in the future) long-term trends.

## 1   Introduction

Ozone ($O_3$) is an important atmospheric trace species: about 90% of the $O_3$ abundance is in the stratosphere, where it protects life on the Earth's surface from harmful ultraviolet (UV) rays from the sun in the stratosphere (IPCC, 2013). The main source of stratospheric $O_3$ is a photochemical process involving oxygen, the so-called Chapman cycle (Langematz, 2019). The stratospheric $O_3$ was observed to decrease since the 1970s, and it was found that this depletion is highly related to the release of chlorofluorocarbons and other halocarbons by mankind (Molina and Rowland, 1974; Montzka et al., 1996). Therefore, 27 nations around the world signed the Montreal Protocol in 1987 to control the emissions of the ozone-depleting species (Murdoch and Sandler, 1997). However, Montzka et al. (2018) monitored an unexpected and persistent increase in global emissions of trichlorofluoromethane (CFC-11) since 2017, and Rigby et al. (2019) pointed out that the increase in CFC-11 emission is attributed to eastern China. Lickley et al. (2020) recently found that CFC-11 and dichlorodifluoromethane (CFC-12) leaking out of old cooling equipment and from building insulation are much larger than had been estimated. Therefore, it is very important the continue the monitoring of ozone all over the world. The remaining ∼10% amount of $O_3$ is located in the troposphere, where it is a pollutant gas that is produced, among others, from interactions with nitrogen oxides and volatile organic compounds (Monks et al., 2015). In addition, the $O_3$ in the free troposphere is also an important greenhouse gas (IPCC, 2013). Xianghe (39.75 °N, 116.96 °E, 50 m a.s.l.), a site located about 50 km east to Beijing, is in a polluted region in North China, with large anthropogenic emissions for $O_3$ precursor gases: carbon monoxide, nitrogen oxides, non-methane volatile organic compounds and methane (European Commission, 2013). Previous studies found that the tropospheric $O_3$ concentrations around Beijing increase significantly since 2002 (Wang et al., 2012; Zhang et al., 2014a; Ma et al., 2016). The high tropospheric $O_3$ concentration has become a serious air pollutant in China, and the tropospheric $O_3$ level in 2015 led to a noticeable increased premature mortality of 0.9% (Feng et al., 2019).

The ground-based Fourier-transform infrared (FTIR) solar absorption spectrometry is a well-established remote sensing technique, which measures an ever-increasing list of chemical compounds along the entire line-of-sight between the ground-based instrument and the sun, thus providing information about the total column as well as the vertical profile for some species, on both short and very long time scales. Within the Network for the Detection of Atmospheric Composition Change - Infrared Working Group (NDACC-IRWG), $O_3$ is an important target gas (De Mazière et al., 2018), and there are about 20 active FTIR sites around the world providing ongoing $O_3$ measurements (http://www.ndacc.org/). The $O_3$ retrieval strategy has been harmonized within NDACC (Vigouroux et al., 2015) and uses the absorption around 1000 cm$^{-1}$ from the spectra recorded with a mercury cadmium telluride (MCT) detector. As NDACC provides long-time series of $O_3$ measurements with high accuracy and precision, these data are used to understand the atmospheric $O_3$ trend (Vigouroux et al., 2015; Steinbrecht et al., 2017), and to validate the satellite measurements (Boynard et al., 2018).



A ground-based FTIR spectrometer (Bruker IFS 125HR) has been installed in June 2018 at Xianghe (39.75°N, 116.96°E; 50 m a.s.l.) to measure the atmospheric carbon dioxide, methane and carbon monoxide (Yang et al., 2019). The FTIR instrument at Xianghe is operated with indium gallium arsenide (InGaAs) and indium antimonide (InSb) detectors, recording the spectra with a spectral range from 1800 to 11000 cm$^{-1}$. Therefore, the NDACC standard O$_3$ retrieval strategy cannot be applied directly

to the Xianghe spectra. Several other infrared microwindows, which have been applied to retrieve O$_3$ from the ground-based FTIR spectra: Lindenmaier et al. (2010) summarized all the related FTIR O$_3$ studies, and it appears that the 3040 cm$^{-1}$ range is often used within the ground-based FTIR community. Takele Kenea et al. (2013) used six micro-windows in the spectral range of 3039.37-3051.90 cm$^{-1}$ for the O$_3$ retrieval at Addis Ababa, Ethiopia. García et al. (2014) tested O$_3$ retrievals in both 3040 and 4030 cm$^{-1}$ ranges at Izaña, Spain, and they found that the precision of O$_3$ total column retrievals from the 3040

cm$^{-1}$ range is 2%, which is much better than the 5% precision obtained in the 4030 cm$^{-1}$ range. However, they found that the total column of O$_3$ from the 3040 cm$^{-1}$ range is about 7% smaller than that retrieved in the standard NDACC 1000 cm$^{-1}$ range.

The aim of this paper is to study the FTIR O$_3$ retrieval in the 3040 cm$^{-1}$ spectral range at Xianghe, and to evaluate the retrieval uncertainty. Section 2 presents the retrieval strategy and the characteristics of the FTIR O$_3$ retrieval at Xianghe. After

that, we show the time series and seasonal variations of FTIR O$_3$ retrievals between June 2018 and December 2019. In section 3, the same retrieval strategy is applied to Maïdo, Reunion Island (21.08 °N, 55.38 °E; 2155 m a.s.l.), which is a NDACC-IRWG affiliated instrument. At both sites, we compare the FTIR O$_3$ measurements with the nearby ozonesonde measurements and the co-located TROPOspheric Monitoring Instrument (TROPOMI) satellite measurements. In addition, the FTIR O$_3$ retrievals (3040 cm$^{-1}$) are compared to standard NDACC FTIR O$_3$ retrievals (1000 cm$^{-1}$) at Maïdo. Finally, the conclusions are drawn

in Section 4.

## 2 FTIR O$_3$ retrievals at Xianghe

The FTIR site at Xianghe Observatory of Whole Atmosphere is operated by the Institute of Atmospheric Physics (IAP), the Chinese Academy of Sciences (CAS). The FTIR system includes a Bruker IFS 125HR instrument, an automatic weather station and a sun tracker system (Yang et al., 2019). The spectra suitable for O$_3$ retrievals are recorded with a maximum optical path

difference of 180 cm, corresponding to a spectral resolution of 0.005 cm$^{-1}$. One specific optical bandpass filter (2000 - 4000 cm$^{-1}$) is inserted in front of the InSb detector in order to improve the signal-to-noise (SNR). The mean SNR of the spectra used in this study is about 1400.

### 2.1 Retrieval strategy

The SFIT4_v9.4.4 algorithm (Pougatchev et al., 1995) is applied to retrieve the O$_3$ profile using the optimal estimation method

(OEM) (Rodgers, 2000)

$$\boldsymbol{x}_r = \boldsymbol{x}_a + \mathbf{A}(\boldsymbol{x}_t - \boldsymbol{x}_a) + \boldsymbol{\epsilon}, \tag{1}$$



where $x_r$, $x_a$ and $x_t$ are retrieved, a priori and true state vectors (all retrieved parameters) and $\mathbf{A}$ is the averaging kernel, representing the sensitivity of the retrieved parameters to the true status. The SFIT4 algorithm minimizes the cost function $(J(x))$

$$J(x) = [y - F(x)]^T \mathbf{S}_\epsilon^{-1} [y - F(x)] + [x - x_a]^T \mathbf{S}_a^{-1} [x - x_a], \tag{2}$$

where $y$ and $F(x)$ are the observed and fitted spectra, respectively, $\mathbf{S}_\epsilon$ is the measurement covariance matrix and $\mathbf{S}_a$ is the a priori covariance matrix. $J(x)$ is the combination of the measurement information and the a priori information, with their weightings determined by $\mathbf{S}_\epsilon$ and $\mathbf{S}_a$. $\mathbf{S}_\epsilon$ is derived from the SNR of the spectra, with its diagonal values set to $1/\text{SNR}^2$ and off-diagonal values to 0. $\mathbf{S}_a$ is derived from the covariance matrix of the Whole Atmosphere Community Climate Model (WACCM) v6 $O_3$ monthly means between 1980 and 2020. The square root of the diagonal elements of $\mathbf{S}_a$ are about 3% near

the surface, 2% in the troposphere, 2.5% in the stratosphere and 1% above the stratosphere.

Table 1 lists the parameters adopted in the retrieval strategy for the FTIR $O_3$ measurements at Xianghe in this study. We selected three retrieval windows (3039.9 - 3040.6 cm$^{-1}$, 3041.5 - 3042.25 cm$^{-1}$ and 3044.7 - 3045.54 cm$^{-1}$) in this study, where the latter two windows are taken from the study of García et al. (2014); the first window has the strongest $O_3$ absorption lines and the least interference with $H_2O$. Comparing to the retrieval windows used in García et al. (2014), the FTIR $O_3$

retrieved total columns from the three windows in this study are less affected by $H_2O$ abundances, and the $O_3$ retrieved profiles are less oscillating by comparison with ozonesonde profiles at Xianghe and Maïdo. For the spectroscopic data, we use the atmospheric line list ATM2019 (https://mark4sun.jpl.nasa.gov/pseudo.html; last access: 26 March 2019). Figure 1 shows an example of the absorption lines and residuals in the three retrieval windows at Xianghe. The root mean square (RMS) of the residual is about 0.2%. It contains a few badly fitted absorptions at $O_3$ line positions in these 3 windows, caused by uncertainties

in the spectroscopy. Further investigations are needed to improve the spectroscopic parameters in this spectral range, but that is beyond the scope of this study. To reduce the influence from the interfering species, $CH_4$, $HCl$, $H_2^{18}O$, $H_2^{17}O$, $H_2O$, $HDO$ and $CO_2$ columns are retrieved simultaneously with the $O_3$ profile. Note that the $H_2O$ isotopes ($H_2^{18}O$, $H_2^{17}O$ and $HDO$) are treated as individual species in the SFIT4 algorithm. The instrument line shape (ILS) is part of the state vector and retrieved simultaneously along with the $O_3$ profile, with an ideal ILS being applied as the a priori input.

The temperature, pressure and $H_2O$ profiles are from the National Centers for Environmental Prediction (NCEP) 6-hourly re-analysis data. For the a priori profiles of $O_3$ and other interfering species, we use the mean of the WACCM model data between 1980 and 2020. Since the broadening effect of absorption lines is related to the pressure and temperature, we can obtain limited vertical information of $O_3$ by fitting the spectra. Figure 2 shows an example of the typical averaging kernel of the FTIR $O_3$ retrieval at Xianghe. The retrieved $O_3$ profile is mainly sensitive to the vertical range between 5 and 40 km. The

degree of freedom for signal (DOFS) is $2.4 \pm 0.3$ ($1\sigma$), indicating that there are two individual pieces of information: partial columns between the surface and 20 km, and between 20 and 40 km. Note that the lower partial column (surface-20 km) is mainly sensitive to the upper troposphere and lower stratosphere (UTLS), and less sensitive to the boundary layer.





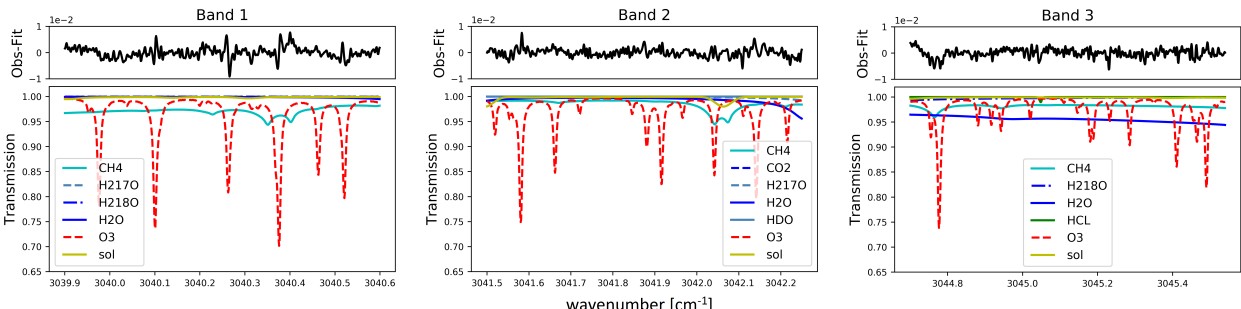

**Figure 1.** Example of spectral fits in the three microwindows for O₃ retrievals at Xianghe. Lower panels: the normalized transmittance from each atmospheric species and solar lines. Upper panels: the difference between the observed and fitted spectra (Obs-Fit).

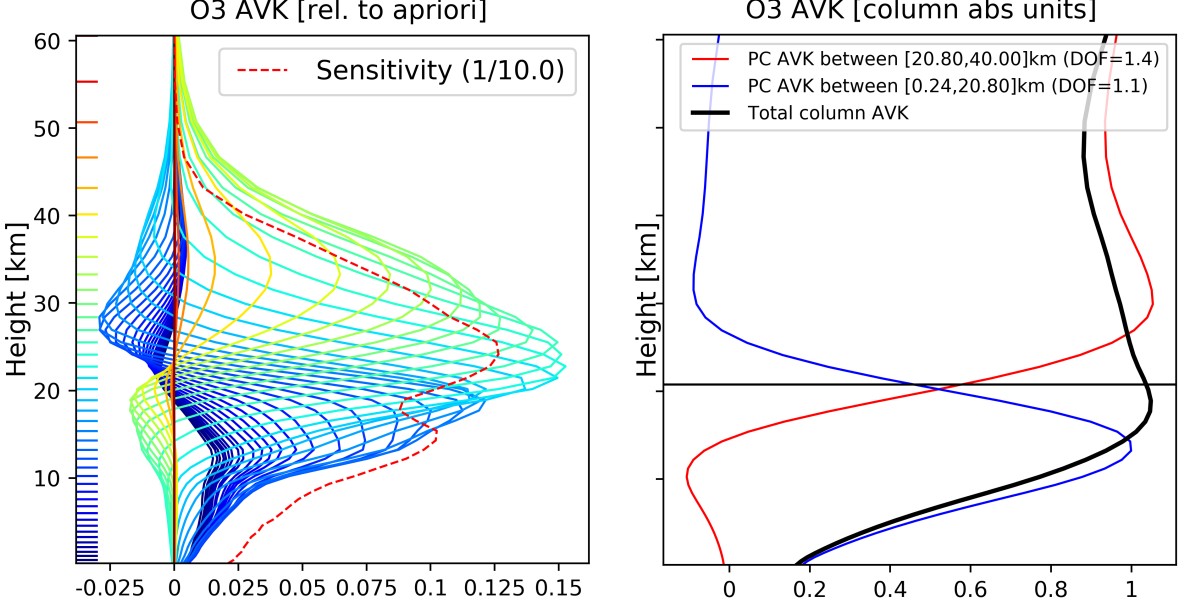

**Figure 2.** Typical vertical sensitivity of the O₃ retrieval at Xianghe. Left panel: the averaging kernel (AVK) matrix whose rows are color coded with the altitude of the retrieval grid (48 layers from the surface to the top of atmosphere). The red dashed line is the sensitivity curve (sum of averaging kernel rows) scaled by 1/10 to bring it to the same scale as the averaging kernel. Right panel: the total column averaging kernel (black) and the partial column (PC) averaging kernels of two individual layers (surface-20 km and 20-40 km) with DOFS equal to 1.1 and 1.4, respectively.





**Table 1.** The retrieval strategy of the FTIR $O_3$ using 3040 cm$^{-1}$ spectral range at Xianghe.

| | |
|---|---|
| Window (cm$^{-1}$) | 3039.9 - 3040.6, 3041.5 - 3042.25 and 3044.7 - 3045.54 |
| Profile retrieval | $O_3$ |
| Column retrieval | $CH_4$, HCl, $H_2^{18}O$, $H_2^{17}O$, $H_2O$, HDO and $CO_2$ |
| Spectroscopy | ATM2019 |
| A priori profile | NCEP and WACCM |
| ILS | polynomial fitting with an ideal ILS as a priori input |
| SNR | ~1400 |
| DOFS | $2.4 \pm 0.3$ |

## 2.2 Uncertainty estimation

According to Rodgers (2000), the error ($\epsilon_r = \boldsymbol{x_r} - \boldsymbol{x_t}$) of the retrieved $O_3$ profile is

$$\boldsymbol{\epsilon_r} = (\mathbf{A} - \mathbf{I})(\boldsymbol{x_t} - \boldsymbol{x_a}) + \mathbf{G}_y \mathbf{K}_b(\boldsymbol{b_t} - \boldsymbol{b}) + \mathbf{G}_y \boldsymbol{\epsilon}_m \qquad (3)$$

where $\boldsymbol{b_t}$ and $\boldsymbol{b}$ are the true and used model parameters, e. g. solar zenith angle (SZA), spectroscopy, temperature; $\mathbf{I}$ is the
unit matrix; $\mathbf{G}_y$ is the contribution matrix; $\mathbf{K}_b$ is the Jacobian matrix for the model parameters; $\boldsymbol{\epsilon}_m$ is the noise of the spectra.
The right side of Eq. 3 contains the smoothing error ($(\mathbf{A} - \mathbf{I})(\boldsymbol{x_t} - \boldsymbol{x_a})$), the model parameter error ($\mathbf{G}_y \mathbf{K}_b(\boldsymbol{b_t} - \boldsymbol{b})$) and the
measurement noise ($\mathbf{G}_y \boldsymbol{\epsilon}_m$). For each component, the systematic and random uncertainties are estimated individually. As the
state vector contains the $O_3$ profile, interfering species and other retrieved parameters, the smoothing error ($(\mathbf{A} - \mathbf{I})(\boldsymbol{x_t} - \boldsymbol{x_a})$)
can be divided into three portions (Zhou et al., 2016), corresponding to smoothing (from the $O_3$ profile), interfering species
and retrieval parameters in Table 2.

The $\boldsymbol{\epsilon}_m$ is derived from the SNR. The systematic uncertainties of both $O_3$ and interfering species a priori profiles are set to
10%, and their random uncertainties are derived from the WACCM data. According to the ATM2019 linelist, the systematic
uncertainties of $O_3$ line intensity, air broadening and pressure broadening are 10-20%, 5-10% and 5-10%, respectively. In
this study, we set 15%, 7.5% and 7.5% for the systematic uncertainties of the $O_3$ line intensity, air broadening and pressure
broadening, respectively, and we assume that there are no random uncertainties. The uncertainties for temperature and $H_2O$
are derived from the difference between NCEP reanalysis data and the European Centre for Medium-Range Weather Forecasts
(ECMWF) ERA5 reanalysis data, where the mean difference is set as the systematic uncertainty and the standard deviation
(STD) of the differences is set as the random uncertainty. The systematic uncertainty of the temperature profile is about 0.5 K
for the whole altitude range, and its random uncertainty is about 2 K below 2 km and 1 K above. The random and systematic
uncertainties for SZA are set to 0.5% and 0.1%, respectively. Table 2 shows the resulting total uncertainty on the retrieved $O_3$
total column and two partial columns. The systematic uncertainty is dominated by the uncertainty from the spectroscopy. The
random uncertainty of the total column is 1.4%, which is coming mainly from the SZA and interfering species uncertainties.
The random uncertainty of the lower partial column (surface-20 km) is 3.6%, which comes mainly the smoothing error and





SZA uncertainty. The random uncertainty of the upper partial column (20-40 km) is 2.2%, which comes mainly from the smoothing error and retrieval parameters uncertainties. To check the estimated random uncertainty, we calculated the mean of daily STD for all days with more than 4 measurements (see Table 2). Keep in mind that daily STD still includes the signal of the diurnal variation, therefore, it might be slightly larger the random uncertainty. In general, the STDs of the total column and the two partial columns are close to the estimated uncertainties, indicating that the random uncertainties have been estimated correctly.

**Table 2.** The estimated retrieval uncertainty of the retrieved $O_3$ total column and two partial columns at Xianghe, together with the corresponding means of the daily STD of the FTIR $O_3$ retrievals.

| $O_3$ | Uncertainty sources | Total column | Surface-20 km | 20-40 km |
|---|---|---|---|---|
| Random [%] | Measurement | 0.2 | 0.3 | 0.3 |
| | Temperature | 0.3 | 0.2 | 0.2 |
| | SZA | 1.0 | 1.0 | 1.0 |
| | Retrieval parameters | 0.1 | 0.7 | 1.4 |
| | Interfering species | 0.9 | 1.7 | 0.7 |
| | Smoothing | 0.6 | 2.9 | 1.2 |
| | **Total** | 1.4 | 3.6 | 2.2 |
| | **Daily STD [%]** | 1.3 | 3.7 | 1.5 |
| Systematic [%] | Spectroscopy | 13.6 | 11.8 | 16.1 |
| | Temperature | 1.4 | 1.7 | 1.9 |
| | SZA | 0.2 | 0.2 | 0.2 |
| | Retrieval parameters | 0.1 | 0.7 | 1.4 |
| | Interfering species | 0.1 | 0.3 | 0.1 |
| | Smoothing | 0.2 | 1.2 | 0.1 |
| | **Total** | 13.7 | 12.0 | 16.3 |

## 2.3 Time series and seasonal variations

Figure 3 shows the time series of the total column of FTIR $O_3$ measurements, as well as two partial columns (surface-20 km and 20-40 km) from June 2018 to December 2019 at Xianghe. The seasonal variation is fitted with a periodic third order function ($\sum_{k=1}^{3} (A_{2k-1}\cos(2k\pi t) + A_{2k}\sin(2k\pi t)$, with $t$ in fraction of year) using all the individual measurements. The mean total column is $8.70 \times 10^{18}$ molecules/cm$^2$, and the mean partial columns between the surface and 20 km, and between 20 and 40 km are $3.42 \times 10^{18}$ molecules/cm$^2$ and $5.05 \times 10^{18}$ molecules/cm$^2$, respectively. The lower partial column (surface-20 km) has a minimum in August-September and a maximum in February-April, while the upper partial column (20-40 km) has a minimum in October-December and a maximum in May-July. The peak-to-peak amplitude of the seasonal variation in the



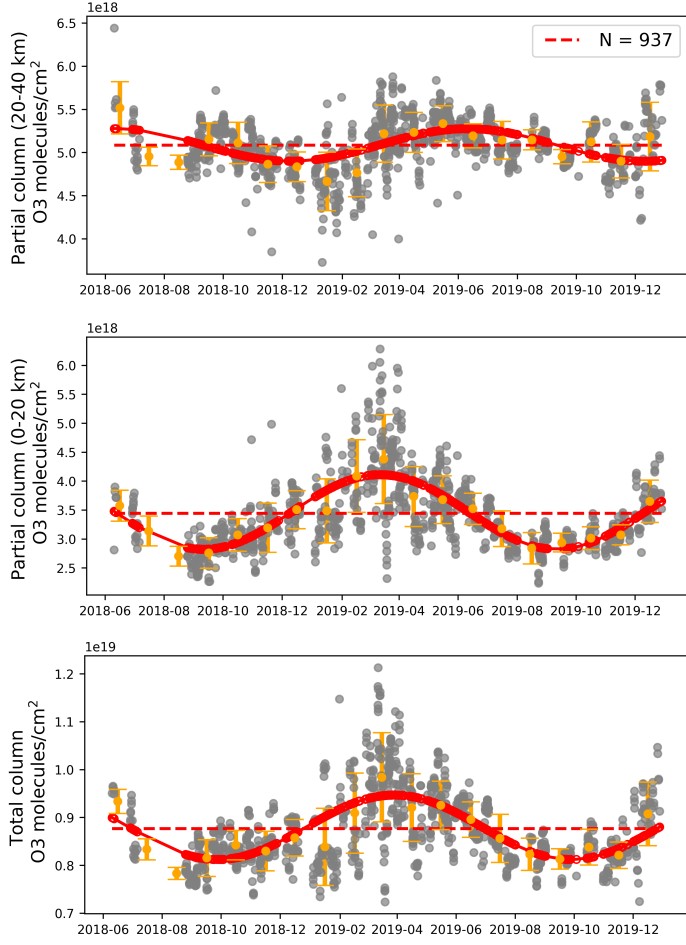

**Figure 3.** The time series of the FTIR retrieved $O_3$ total column (bottom), as well as the partial column between surface and 20 km (middle) and the partial column between 20 and 40 km (top) from June 2018 to December 2019 at Xianghe. The grey dots are each individual retrieval. The orange dots and errorbars are the monthly means and STDs. The red dashed line is the mean and red solid line is the fitted seasonal variation. N is the measurement number.

partial column between surface and 20 km is $1.3 \times 10^{18} \text{molecules/cm}^2$, which is much larger than that in the partial column between 20 and 40 km of $0.4 \times 10^{18} \text{molecules/cm}^2$. Therefore, the seasonal variation of the total column is dominated by the lower partial column (surface-20 km). The FTIR $O_3$ retrieved lower partial column (surface-20 km) has a maximum sensitivity in the UTLS region (see Figure 2). The ozonesonde measurements between 2002 and 2010 at Beijing (Wang et al., 2012) showed that the high $O_3$ concentrations are in the UTLS in later winter and spring with a year-to-year variation, and the low $O_3$ concentrations in the UTLS in August-September. In the middle and upper stratosphere (20-40 km), the maximum observed in summer is mainly due to the higher photochemical production in this season (Perliski et al., 1989).





## 3 Validation of O$_3$ total and partial columns

On the purpose of validating the FTIR O$_3$ retrievals at Xianghe in the 3040 cm$^{-1}$ spectral range, we first compare them with nearby ozonesonde and co-located TROPOMI measurements. Secondly, we apply the same retrieval strategy (3040 cm$^{-1}$) to the FTIR observations at the Maïdo (Reunion Island) which is a NDACC affiliated site, and we compare them with the standard
NDACC O$_3$ retrievals (1000 cm$^{-1}$) at this site, as well as with nearby ozonesonde and co-located TROPOMI measurements.

### 3.1 Ozonesonde

The ozonesonde is a compact, lightweight balloon-borne instrument, which is coupled to a meteorological radiosonde. The balloon is launched at the surface and ascends up to the upper stratosphere (about 35 km), providing in-situ measurements of the ozone profile with a high vertical resolution of about few hundred meters (Thompson et al., 2003). According to Deshler
et al. (2017), the accuracy of the ozonesonde profile is within 10% in the troposphere and 5% in the stratosphere. The precision of the ozonesonde is about 3-5% (Deshler et al., 2008; Liu et al., 2009).

The ozonesondes are launched at Beijing Observatory (39.81°N, 116.47°E, 31 m a.s.l.), about 50 km west of the Xianghe site. The ozonesonde instrument was developed at IAP, CAS (named as IAP ozonesonde). The IAP ozonesonde consists of an anode cell and a cathode cell, and uses an electrochemical method, which is similar to the Electrochemical Concentration Cell
(ECC) type ozonesonde. For the detailed information about instrument, please refer to Zhang et al. (2014b). The performance of the IAP ozonesonde measurements has been evaluated by comparison with other ECC ozonesonde measurements (Zhang et al., 2014b): the average difference in the ozone partial pressure between the IAP and ECC ozonesondes is 0.3 mPa from the surface to 2.5 km, close to zero from 2.5 to 9 km and generally less than 1 mPa for layers higher than 9 km. Note that we have applied the pressure pump efficiency corrections to the IAP ozonesonde (Zheng et al., 2018), resulting in higher ozone
detecting performance relative to the results in Zhang et al. (2014b). The IAP ozonesonde measurements used in this study cover the period between June 2018 and February 2019, after which the ozonesonde measurements stopped.

The ozonesonde data performed at Gillot, Reunion Island (20.89°S, 55.53°E, 8 m a.s.l.) are affiliated to the NDACC (De Mazière et al., 2018) and the Southern Hemisphere Additional Ozonesondes (SHADOZ) network (Thompson et al., 2003). Detailed information about the ozonesonde measurements at Gillot can be found in Thompson et al. (2014); Witte et al. (2017),
where the ozonesonde measurements are applied to understand the tropospheric ozone increases over the southern Africa region. Gillot is about 26 km away from Maïdo, and is considered representative for the ozone concentrations at Maïdo (Duflot et al., 2017). The ozonesonde measurements used in this study cover the period between April 2013 and July 2017.

We select FTIR measurements within a ± 3-hours window around each ozonesonde, and take the averaged FTIR retrieval and the ozonesonde measurement as one FTIR-sonde data pair. In total, we have 16 and 53 data pairs at Xianghe and Maïdo,
respectively. As the vertical resolution of ozonesondes is much higher than that of the FTIR retrievals, the ozonesonde profiles are smoothed with the FTIR averaging kernel to reduce the smoothing error in the comparison between both (Rodgers and Connor, 2003):

$$x_s{}' = x_{F,a} + \mathbf{A}(x_s - x_{F,a}), \tag{4}$$





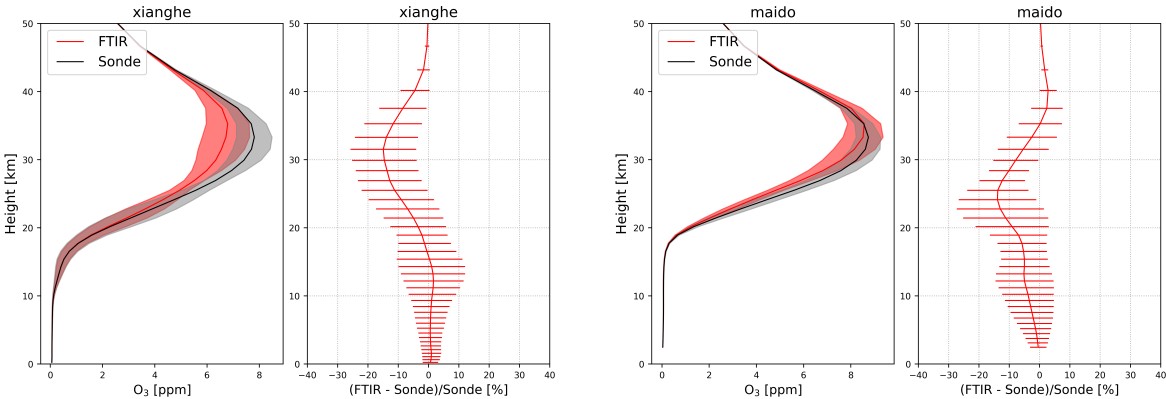

**Figure 4.** The $O_3$ profiles from the smoothed ozonesonde and FTIR retrievals (the solid line is the mean and the shadow is the STD), together with their relative differences (the solid line is the mean and the errorbar is the STD) at Xianghe (left) and Maïdo (right).

where $x_{F,a}$ is the FTIR a priori profile, $x_s$ is the ozonesonde profile, $x_s'$ is the smoothed ozonesonde profile, and $\mathbf{A}$ is the FTIR averaging kernel. To apply the smoothing correction, we have extended the ozonesonde profile to the top of atmosphere using the FTIR a priori profile.

The profiles of the FTIR retrievals and ozonesonde measurements, together with their relative differences at Xianghe and
Maïdo are shown in Figure 4. In general, the relative difference profiles at these two sites are similar: within $\pm15\%$ below 20 km and between -30 % and 10% between 20 km and 40 km. The total column observed by ozonesonde is $6.4 \pm 6.0$ ($1\sigma$) % and $9.0 \pm 4.3$ % larger than the FTIR (3040 cm$^{-1}$) retrievals at Xianghe and Maïdo, respectively. To check the impact of the $O_3$ columns above the maximum height of the ozonesonde, we also compare the FTIR column between the surface to the maximum altitude of each co-located ozonesonde profile, where the ozonesonde measurements are $6.2 \pm 6.1$ % and
$9.7 \pm 7.0$ % larger than the FTIR retrievals at Xianghe and Maïdo, respectively. As a result, the impact of extending the ozonesonde profile to higher altitude with the FTIR a priori profile is relatively small compared to the large uncertainty. The comparisons between the total and partial columns (surface-20 km and 20-40 km) retrieved from the FTIR and the ozonesonde measurements are listed in Table 3.

## 3.2 TROPOMI satellite measurements

The Sentinel-5 Precursor (S5P) satellite, carrying the TROPOMI instrument, was successfully launched into a sun-synchronous orbit on 13 October 2017, providing a high horizontal resolution of $7 \times 3.5\ km^2$ before 6 August 2019 and of $5.5 \times 3.5\ km^2$ since then. TROPOMI observes a number of trace species globally, including $O_3$, with a nadir view. In this section, the TROPOMI offline (OFFL) total ozone column measurements are compared with the FTIR $O_3$ (3040 cm$^{-1}$) retrievals at Xianghe and Maïdo. The pre-launch requirements regarding accuracy and precision of the TROPOMI OFFL $O_3$ total column product are
3.5-5.0% and 1.6-2.5%, respectively. TROPOMI OFFL $O_3$ total column products have been validated by ground-based Brewer,





**Table 3.** The mean and STD (mean/STD) of the relative differences between the FTIR $O_3$ (3040 cm$^{-1}$) retrievals with other datasets (ozonesonde, TROPOMI and FTIR $O_3$ (1000 cm$^{-1}$ retrievals)) in total column and two partial columns (surface-20 km and 20-40 km) at Xianghe and Maïdo. The relative difference is calculated as (FTIR-other)/other × 100%.

|  | Datasets | Xianghe mean/std [%] | Maïdo mean/std [%] |
|---|---|---|---|
| Total column | Ozonesonde | -6.4/6.0 | -9.0/4.3 |
|  | FTIR (1000 cm$^{-1}$) |  | -8.4/1.1 |
|  | TROPOMI | -5.5/2.0 | -6.1/1.3 |
| Surface-20 km | Ozonesonde | -0.3/6.0 | -2.4/6.6 |
|  | FTIR (1000 cm$^{-1}$) |  | 0.8/4.4 |
| 20-40 km | Ozonesonde | -10.3/7.8 | -10.1/6.5 |
|  | FTIR (1000 cm$^{-1}$) |  | -10.8/1.8 |

Dobson and Zenith Scattered Light-Differential Optical Absorption Spectroscopy (ZSL-DOAS) measurements. It is found that the mean bias between the TROPOMI and ground-based measurements is +0.1% and the STD of the relative differences is about 2.0%, which is within the mission requirements (Garane et al., 2019).

We select TROPOMI satellite OFFL data within a ± 6 hours temporal window and within a ± 1.0 ° latitude and ± 3.0

5 ° longitude box of each FTIR $O_3$ measurement at Xianghe and Maïdo. As the FTIR measurements at Xianghe start in June 2018, in this study, we compare the FTIR measurements with TROPOMI OFFL data between June 2018 and December 2019 at both sites. As mentioned in Section 2.1, the FTIR a priori profile is derived from the WACCM model, while the a priori profile of the TROPOMI retrieval is from a column-classified ozone profile climatology (Heue et al., 2018). In order to reduce the influence of different a priori profiles, we substitute the satellite a priori profile for the ground-based FTIR a priori profile

10 when comparing both datasets

$$\boldsymbol{x}'_r = \boldsymbol{x}_r + (\mathbf{I} - \mathbf{A})(\boldsymbol{x}_{s,a} - \boldsymbol{x}_{F,a}), \tag{5}$$

where $\boldsymbol{x}'_r$ is the adapted FTIR profile by using satellite a priori profile as the a priori profile; $\boldsymbol{x}_r$ is the original FTIR retrieved profile; $\boldsymbol{x}_{s,a}$ and $\boldsymbol{x}_{F,a}$ are the satellite and FTIR a priori profiles. TROPOMI provides the column averaging kernel ($\boldsymbol{A_s}$) together with the total column, therefore, we applied the smoothing correction to the adapted FTIR profile:

15 $$TC'_r = TC_{s,a} + \boldsymbol{A_s}\boldsymbol{PC_{dry,air}}(\boldsymbol{x}'_r - \boldsymbol{x}_{s,a}), \tag{6}$$

where $TC_{s,a}$ is the TROPOMI a priori total column and $TC'_r$ is the FTIR retrieved total column after a priori profile substitution and taking TROPOMI vertical sensitivity into account.

Figure 5 shows the time series of the co-located FTIR and TROPOMI $O_3$ total columns, together with their differences and correlations at Maïdo and Xianghe. Similar to ozonesonde measurements, the TROPOMI measurements are 5.5 ± 2.0 % and



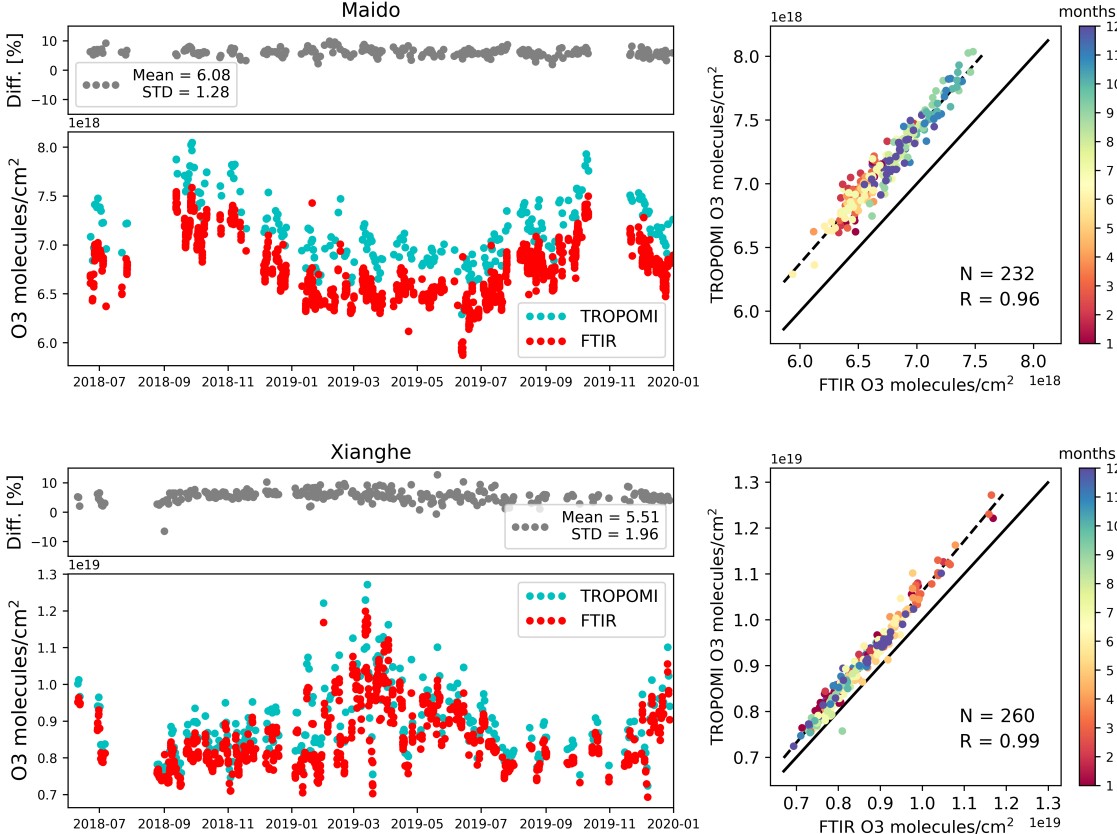

**Figure 5.** The time series of the co-located total columns from FTIR and TROPOMI measurements, together with their relative differences ((FTIR - TROPOMI)/TROPOMI × 100 %) and their correlations at Maïdo and Xianghe between June 2018 and December 2019. N is the co-located number of data pairs and R is the correlation coefficient. The correlation dots are coloured with their measurement months. The black dashed line is the linear regression line.

6.1 ± 1.3 % larger than the FTIR (3040 cm$^{-1}$) total columns at Xianghe and Maïdo, respectively. In addition, there is no clear time dependence in the relative differences between FTIR and TROPOMI total columns.

There is a good correlation between the FTIR and TROPOMI measurements at Xianghe (R=0.99) and Maïdo (R=0.96). The seasonal and synoptic variations (phase and amplitude) of total columns of O$_3$ from the FTIR and TROPOMI measurements are very close to each other at both sites. As an example, FTIR and TROPOMI measurements show that there is a large enhancement of O$_3$ total column on 31 January 2019 at Xianghe (see Figure 6). Keep in mind that we should focus on the total column and two partial columns of FTIR measurements instead of the FTIR retrieved O$_3$ profile due to its limited vertical information. According to the FTIR measurements, both partial columns increase on that day, but the large increase of the total column mainly results from the enhancement of the lower partial column from the surface to 20 km altitude. There is



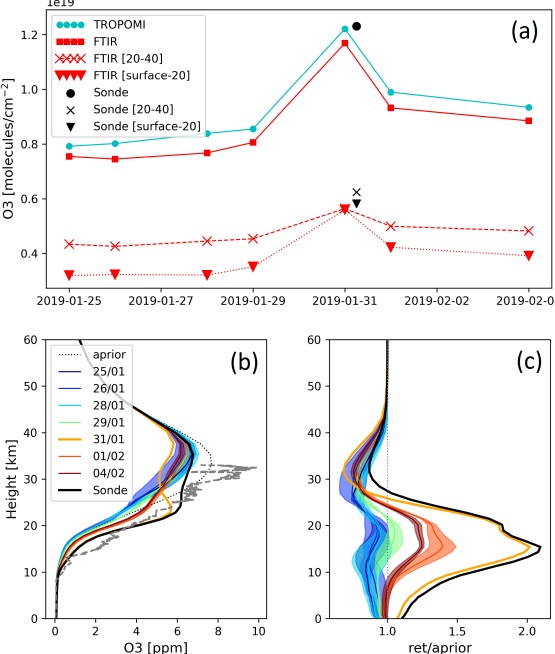

**Figure 6.** (a): the time series of the total and partial columns daily means from co-located FTIR and TROPOMI measurements between 25 January 2019 and 4 February 2019, and one ozonesonde profile (with smoothing using FTIR averaging kernel), on 31 January 2019 at Xianghe. For visualizing, the ozonesonde measurement is shifted by 6 hours. (b): the FTIR a priori (grey dot line) and retrieved (colored with date) profiles during this period. The shadow is the STD of the retrieved profile for each day. The grey dashed line is the original ozonesonde profile, and the back solid line is the smoothed ozonesonde profile. (c): the ratios of the FTIR retrieved profiles and the smoothed ozonesonde profile to the FTIR a priori profile.

one ozonesonde profile available on 31 January 2019, which confirms that the $O_3$ mole fraction is much larger compared to the FTIR a priori profile above 10 km, especially in the UTLS region. The smoothed ozonesonde profile is close to the FTIR retrieved profile below 23 km, which is consistent with our results in Table 3.

### 3.3 FTIR (1000 cm$^{-1}$) retrievals

5 Maïdo is an NDACC station, where FTIR measurements using a MCT detector are carried out (Baray et al., 2013; Zhou et al., 2018). The harmonized $O_3$ standard retrieval strategy using 1000-1005 cm$^{-1}$ has been performed at Maïdo, so that we can compare the FTIR (3040 cm$^{-1}$) with the FTIR (1000 cm$^{-1}$) retrievals for total column as well as for two partial columns. The FTIR $O_3$ (1000 cm$^{-1}$) retrieval has a DOFS of about 4 to 5, because in this spectral range it benefits from more $O_3$ lines with different intensities. The systematic and random uncertainties of the total column from FTIR $O_3$ (1000 cm$^{-1}$) retrievals are

10 about 3.0% and 1.0%, respectively. The systematic and random uncertainties of the surface to 20 km partial column retrievals are about 3.2% and 2.5%, respectively, and of the 20 to 40 km partial column retrievals are about 3.4% and 1.5%, respectively.



Both precision and accuracy are better using $O_3$ (1000 $cm^{-1}$) than $O_3$ (3040 $cm^{-1}$), which explains why the MCT spectral region is preferred at NDACC stations where these measurements are available. The systematic uncertainty is also dominated by the spectroscopy (HITRAN2008; Rothman et al. (2009)), where we set 3% for the uncertainty of line intensity (NDACC-IRWG recommendation, based on total column comparisons with Dobson and Brewer measurements, e.g. in Vigouroux et al.

5 (2008)).

The a priori profiles of the FTIR $O_3$ (1000 $cm^{-1}$) retrievals are the same as those of the FTIR $O_3$ (3040 $cm^{-1}$) retrievals (see Section 2.1). To take the low vertical resolution of the FTIR (3040 $cm^{-1}$) retrieval into account, the FTIR (1000 $cm^{-1}$) retrieved profile is smoothed with the FTIR (3040 $cm^{-1}$) averaging kernel

$$x_{1000}' = x_a + \mathbf{A}(x_{1000} - x_a), \tag{7}$$

where $x_a$ is the FTIR a priori profile; $x_{1000}$ is the FTIR (1000 $cm^{-1}$) retrieved profile, $x_{1000}'$ is the FTIR (1000 $cm^{-1}$) retrieved profile after smoothing with the FTIR (3040 $cm^{-1}$) averaging kernel ($\mathbf{A}$).

The time series of the hourly means retrieved $O_3$ total column and partial columns (surface-20 km and 20-40 km) in the 3040 $cm^{-1}$ and 1000 $cm^{-1}$ spectral ranges, together with their differences and correlations are shown in Figure 7. Both $O_3$ datasets show the same seasonal variations in the total column and the two partial columns. The mean and STD of their relative

differences are also listed in Table 3. The $O_3$ (3040 $cm^{-1}$) total columns have a negative bias of $8.4 \pm 1.1$ % compared to the $O_3$ (1000 $cm^{-1}$) total columns. For the lower partial column (surface-20 km), the two FTIR $O_3$ retrievals are close to each other, with a mean relative difference of $0.8 \pm 4.4$ %. The $O_3$ upper partial column (20-40 km) retrieved in the 3040 $cm^{-1}$ spectral range is $10.8 \pm 1.8$ % smaller than the one retrieved in the 1000 $cm^{-1}$ spectral range.

García et al. (2014) found that there is an underestimation of 7% in the FTIR $O_3$ (3040 $cm^{-1}$) total column compared to

FTIR $O_3$ (1000 $cm^{-1}$) retrievals at Izaña based on the HITRAN2012 spectroscopy (Rothman et al., 2013), which is generally in good agreement with our result ($8.4 \pm 1.1$ %) at Maïdo. In this study, we also looked at comparisons between the two partial columns. The biases observed between FTIR $O_3$ (3040 $cm^{-1}$) and FTIR $O_3$ (1000 $cm^{-1}$) on one hand and between FTIR $O_3$ (3040 $cm^{-1}$) and ozonesondes on the other hand are similar (see Table 3), pointing to an underestimation of the FTIR retrieved total and partial columns products in the 3040 $cm^{-1}$ spectral range; the bias is coming mainly from the 20-40 km

partial column bias.

The FTIR $O_3$ (3040 $cm^{-1}$) retrievals and the FTIR $O_3$ (1000 $cm^{-1}$) retrievals are highly correlated, with R values of 0.95, 0.87 and 0.89 in the total column, the lower partial column (surface-20 km) and the upper partial column (20-40 km), respectively. The mean of daily STDs for the days for which more than 4 measurements are available are 0.57% and 0.58% in the total column, 2.41% and 0.85% in the lower partial column (surface-20 km), and 0.77% and 0.71% in the upper partial

column (20-40 km) for FTIR $O_3$ (3040 $cm^{-1}$) and smoothed FTIR $O_3$(1000 $cm^{-1}$) retrievals, respectively. In summary, the two FTIR $O_3$ retrievals at Maïdo show a similar precision in the total column and the upper partial column (20-40 km), while the FTIR $O_3$ lower partial columns (surface-20 km) retrievals are more variable in the 3040 $cm^{-1}$ than in the 1000 $cm^{-1}$ spectral range.

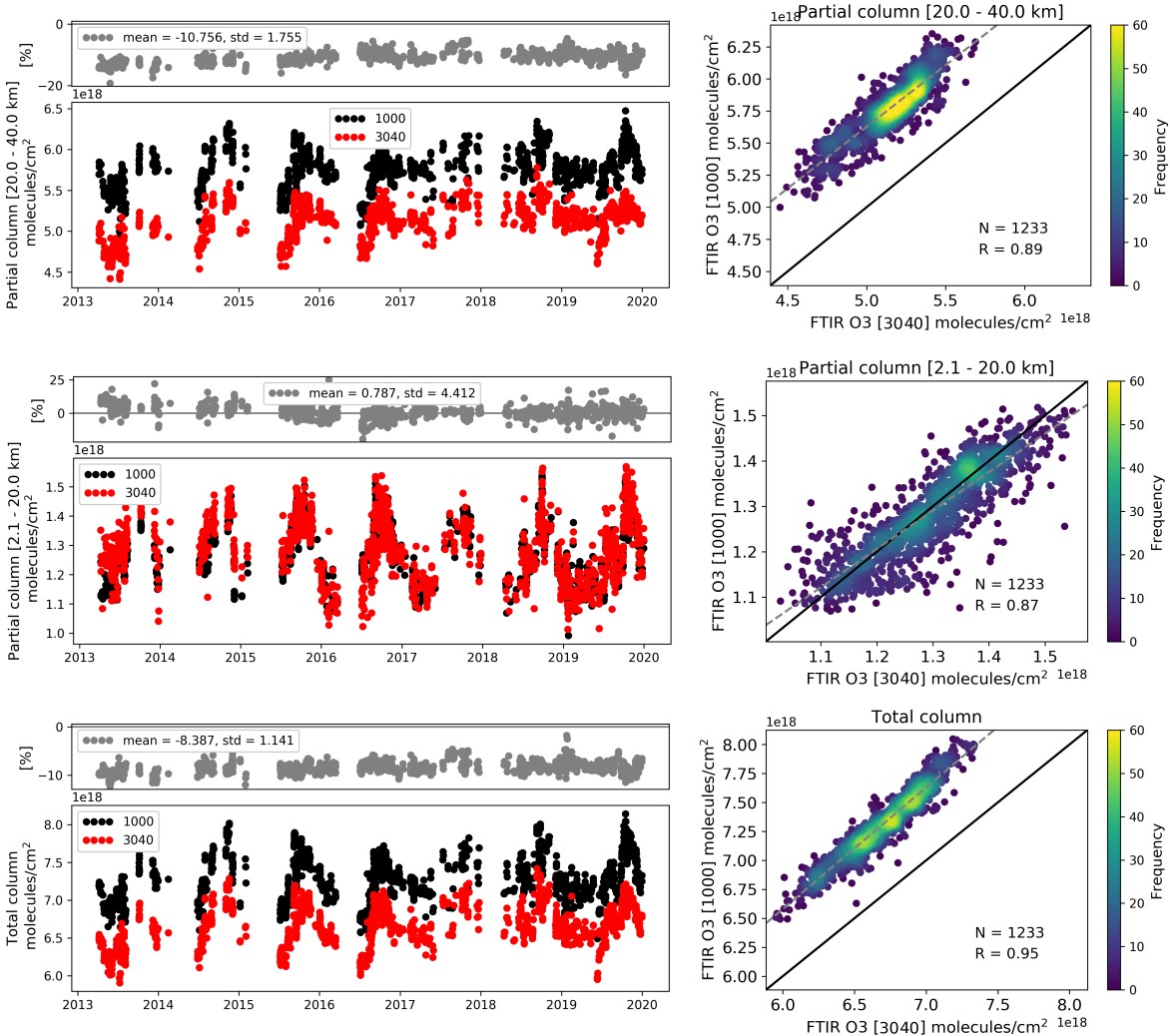

**Figure 7.** The time series of the co-located hourly means of total columns (bottom), partial columns between surface and 20 km (middle) and partial columns between 20 and 40 km (top) from FTIR retrievals using the 3040 cm$^{-1}$ (red) and the 1000 cm$^{-1}$ (black) spectral ranges, together with their relative differences ((FTIR_3040 - FTIR_1000)/FTIR_1000 × 100 %) and their correlations, in which color intensity corresponds to data frequency, at Maïdo between 2013 and 2019. The grey dashed line is the linear regression line. N is the co-located number of data pairs and R is the correlation coefficient.





# 4    Conclusions

The standard NDACC-IRWG $O_3$ retrieval uses the retrieval window of 1000-1005 cm$^{-1}$ recorded with a MCT detector. However at some ground-based atmospheric observatories the FTIR solar absorption instruments are not configured for operation with a MCT detector. This is the case at Xianghe, China (39.75 °N, 116.96 °E), where the FTIR instrument is operated with

InSb and InGaAs detectors covering the spectral range from 1800 cm$^{-1}$ to 11000 cm$^{-1}$. Therefore, in this paper, we present ground-based FTIR $O_3$ retrievals at Xianghe between June 2018 and December 2019 using the standard NDACC-IRWG SFIT4 v9.4.4 retrieval algorithm and the spectral windows (3039.9 - 3040.6 cm$^{-1}$, 3041.5 - 3042.25 cm$^{-1}$ and 3044.7 - 3045.54 cm$^{-1}$). The resulting averaging kernel shows that the retrieved $O_3$ profile is mainly sensitive to the vertical range between 5 and 40 km, and the DOFS is 2.4±0.3 (1$\sigma$), indicating that we can retrieve two independent partial columns, one from the

surface to 20 km and a second one from 20 to 40 km altitude. Based on the optimal estimation method, we have estimated the systematic and random uncertainties of the retrieved FTIR $O_3$ total columns to be about 13.6% and 1.4%, respectively, in which the random error is generally in good agreement with the observed daily STD of the FTIR retrievals.

The FTIR retrieval systematic uncertainty is then verified by comparing the FTIR $O_3$ retrievals in the 3040 cm$^{-1}$ spectral range with nearby ozonesonde and co-located TROPOMI measurements at Xianghe and Maïdo, and with NDACC standard

FTIR $O_3$ (1000 cm$^{-1}$) retrievals at Maïdo. There is a systematic underestimation by 5.5-9.0% in the FTIR $O_3$ (3040 cm$^{-1}$) total column retrievals, which is within the estimated systematic uncertainty and mainly due to the spectroscopic uncertainties. According to ozonesonde measurements and standard NDACC FTIR $O_3$ retrievals, the underestimation of the FTIR (3040 cm$^{-1}$) $O_3$ total column mainly results from the underestimation by 10.1-10.8% in the upper partial column (20-40 km). The systematic uncertainty is relatively small in the lower partial column (surface-20 km), which is within 2.4%.

At Xianghe, the FTIR retrieved $O_3$ partial columns between surface and 20 km show a maximum in February-April and a minimum in August-September, with a peak-to-peak amplitude of $1.3 \times 10^{18}$molecules/cm$^2$, while the 20-40 km partial columns show a maximum in May-July and a minimum in October-December, with a peak-to-peak amplitude of $0.4 \times 10^{18}$molecules/cm$^2$. As the amplitude of the seasonal variation in the lower partial column (surface-20 km) is much larger than the one in the upper partial column (20-40 km), the seasonal variation of the total column is dominated by the lower

partial column. The FTIR (3040 cm$^{-1}$) retrievals at Xianghe and Maïdo show the same seasonal and synoptic $O_3$ variations as seen by the TROPOMI satellite measurements and the NDACC standard FTIR $O_3$ retrievals at Maïdo.

The ongoing FTIR $O_3$ total and partial columns (surface-20 km and 20-40 km) data at Xianghe can provide useful information on $O_3$ synoptic and seasonal variations and long-term trends. Based on the successful and consistent $O_3$ retrieved results at Xianghe and Maïdo, the retrieval strategy used in this study can be extended to other FTIR sites recording the 3040 cm$^{-1}$

spectral range.

*Data availability.*    The NDACC standard FTIR $O_3$ (1000 cm$^{-1}$) retrievals at Maïdo and the ozonesonde measurements are publicly available at the NDACC archive (ftp://ftp.cpc.ncep.noaa.gov/ndacc/; last access: 20 January 2019). The TROPOMI off-line data are publicly available





at ESA Copernicus Open Access Hub (https://scihub.copernicus.eu/). The ozonesonde and FTIR O$_3$ (3040 cm$^{-1}$) retrievals at Xianghe used in this study can be obtained by contacting the authors.

*Competing interests.* The authors declare that they have no conflict of interest.

*Acknowledgements.* This research was funded by the National Key R&D Program of China (Nos. 2017YFB0504000 and 2017YFC1501701)
5  and the National Natural Science Foundation of China (41975035). The FTIR site at Reunion Island are operated by the BIRA-IASB and locally supported by LACy/UMR8105, Université de La Réunion. We would like to thank Weidong Nan, Qun Cheng and Rongshi Zou (IAP) for the FTIR maintenance at Xianghe, Geoffrey C. Toon (JPL, NASA) for sharing the ATM2019 spectroscopy, Daan Hubert (BIRA-IASB) for useful discussion about the ozonesonde measurements, and Ball William (ETH zürich) for useful discussions. We also acknowledge the NDACC-IRWG network for providing the retrieval code and data, and ESA for providing the TROPOMI products. The work done by MZ
10 and BIRA colleagues has been supported through the Copernicus Atmospheric Monitoring Service contracts (CAMS-84 and CAMS-27).

*Author contributions.* MZ wrote the manuscript. MZ and PW designed the experiment, with the significant inputs from BL, CV, LR, MDM. CH, NK, TW, YY, DJ, JMM collected the FTIR measurements at Xianghe and Maïdo. JZ, YX, HC, VD, FP provided and studied the ozonesonde measurements at Beijing and Gillot. All the authors read and commented on the manuscript.





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
