# Peer review of "Ground-based FTIR $O_3$ retrievals from the 3040 cm-1 spectral range at Xianghe, China"

_Atmospheric Measurement Techniques, 2020_

## Referee Comment (RC1) · Anonymous Referee #1 · 27 May 2020

General comments: This manuscript presents a generally well written study on ozone retrievals from ground-based Fourier-transform infrared (FTIR) solar absorption spectra in the 3040 cm-1 spectral region. This is not a new approach, it was previously used (Rinsland et al., 1996; Goldman et al., 1999; Meier et al., 2005; Fu et al., 2007; Sung et al., 2007) for ozone retrievals, and as the study shows, this is not an optimal region for retrieving ozone. The 1000 cm-1 region proved to be more adequate for this purpose (Lindenmaier et al., 2010), and was adopted by the Network for the Detection of Atmospheric Composition Change for its harmonized FTIR ozone retrieval strategy (Vigouroux et al., 2015). However, for this particular site, Xianghe, where the spectral range is limited to the 1800 - 11000 cm-1 domain, it can give useful information about the seasonal ozone variations and its long-term trends. This retrieval approach can be

extended to other FTIR sites recording spectra in this range. Therefore, I recommend this study for publication in AMT after minor revisions.

1) P3L25 – You mention "One specific optical bandpass filter…." Can you please be more specific? Is that the standard narrow bandpass Filter 3 (2420-3080 cm-1) used by the NDACC-IRWG community? Is it wedged?

2) P4L2 – Explain what is epsilon.

3) P4L12-14 – What was the criteria for choosing these three particular windows for your retrievals? In the 2000 – 4000 cm-1 there are other windows that could be used for ozone retrievals, e.g. 2775 cm-1, 3023 cm-1. Also, you affirm that the first window has the strongest ozone absorption lines and the least interference with H2O. Why did you add the other two? Wouldn't have been enough to use only the first? You should explain in the text. Also, clarify if these windows were used simultaneously.

4) P4L19-20 – Have you tried fitting the minor interfering species to improve the residual? For example, for the 3039.9 – 3040.6 cm-1 window, what is the result if you fit also CH3Cl? Solar lines are not mentioned at all in the text, only in the caption of Figure 1. Among the weak species for this same window, beside solar lines you have HDO, NH3, and OH. Have you tried fitting these species? It would be great to add some text here and explain how you picked the interfering species for each window rather than just list them.

5) P5 Figure 1 – Please enlarge the panels for each window (make them as those in Figure 2 for clarity. Also, the values on the x and y axes are too small, hard to read. Bring them at the size in Figure 2.

6) P9L15-20 – This part is confusing. What is the accuracy and precision of the IAP ozonsondes? What does "higher ozone detecting performance" mean? I would give numbers here, the error for the IAP ozonsondes.

7) P10 Figure 4 – Please enlarge the numbers on the x and y axes.

8) P11L6 – FTIR measurements are compared with TROPOMI OFFL at both sites, but for what window? Specify.

9) P16L8 – To me it looks like it is more 10 to 40 km rather than 5 and 40 km (text). In my opinion it is not correct then to use surface to 20 km. Use 10 to 20 km in the entire manuscript, I think it is more appropriate.

There are some typos in the text:

P2L15 – Change "the continue" to "to continue"

P6L24 – Change "mainly the" to "mainly from the"

P13 Figure 6 caption L5 – Change "and the back solid line" to "and the black solid line"

References: Rinsland CP, Connor BJ, Jones NB, Boyd I, Matthews WA, Goldman, A, et al. Comparison of infrared and Dobson total ozone columns measured from Lauder, New Zealand. Geophys Res Lett 1996; 23:1025–8.

Goldman A, Paton-Walsh C, Bell W, Toon GC, Blavier JF, Sen, B, et al. Network for the Detection of Stratospheric Change Fourier trans- form infrared intercomparison at Table Mountain Facility, Novem- ber 1996. J Geophys Res 1999; 104:30481–503.

Meier A, Paton-Walsh C, Bell W, Blumenstock T, Hase F, Goldman, A, et al. Evidence of reduced measurement uncertainties from an FTIR instrument intercomparison at Kiruna, Sweden. JQSRT 2005; 96:75–84.

Fu D, Walker KA, Sung K, Boone CD, Soucy MA, Bernath PF. The portable atmospheric research interferometric spectrometer for the infrared, PARIS-IR. JQSRT 2007; 103:362–70.

Sung K, Skelton R, Walker KA, Boone CD, Fu D, Bernath P. N2O and O3 Arctic column amounts from PARIS-IR observations: retrievals, characterization and error analysis. JQSRT 2007; 107:385–406.

Lindenmaier R, Batchelor RL, Strong K, Fast H, Goutail F, Kolonjari F, et al. An evaluation of infrared microwindows for ozone retrievals using the Eureka Bruker 125HR Fourier transform spectrometer, JQSRT 2010; 111(4):569-585.

---

## Referee Comment (RC2) · Anonymous Referee #2 · 8 Jun 2020

General comments:

The authors present a study on ozone retrievals from infrared spectra recorded in Xinghe, China and on Reunion Island. Data from these sites are highly needed since these areas are poorly represented in the networks. This study uses the 3040 cm-1 spectral region and presents results of a one year time series and a characterisation of the 3040 cm-1 ozone product. Moreover, using spectra from Reunion Island data obtained from the 3040 cm-1 region are compared with those with the standard NDACC retrieval at 1000 cm-1. The comparison shows a good correlation, but a bias of 5.5 to 9.0 % and reduced degrees of freedom compared to the standard microwindow. Ozone retrievals in the 3040 cm-1 cm-1 region are very useful since there are several FTIR spectrometers without an MCT detector around the globe.

[Figure]

For the 3040 cm-1 retrieval a modified version of the recipe of Garcia et al., 2014, was used. As a result, the key findings are very similar to those obtained by Garcia et al., 2014. However, since the recent study doesn't use exactly the same recipe, strictly speaking, it cannot be used as confirmation of the Garcia recipe and as an extension including more sites covering different conditions. To my impression, it is not clear whether it is a confirmation of the Garcia paper showing similar retrieval results or whether there is an improvement as compared to the Garcia paper. If the authors claim the latter this should be demonstrated or at least discussed in detail. To do so the authors might think in adding a Garcia type retrieval for comparison.

Therefore, I would recommend publishing this paper after major revisions although the paper is well written and fits well to the scope of AMT. Please also see specific comments below.

Specific comments:

- The statement in the abstract 'as the harmonized . . . uses the 1000 cm-1 spectral range, we designed an alternative O3 retrieval strategy . . .' is not correct since there is a published 'alternative' retrieval recipe for FTIR sites without MCT detector as published by Garcia et al., 2014.

- The recipe from Garcia et al. 2014, has been modified. The modifications made and the rationale behind these modifications should be described in more detail. Moreover, a comparison with retrieval results using the full recipe from Garcia et al. would be very useful to see the effect of these modifications.

- p. 4: 'a few badly fitted absorptions': Fig. 1 shows strong residuals at ozone line positions in particular in microwindow 1, not included in the Garcia paper. Does this additional window really improves the fit results although the line list needs improvement for this window?

- p. 3: 'One specific optical bandpass filter (2000 – 4000 cm-1)': This is not the standard

NDACC type optical filter. The NDACC type filters provide a smaller bandwidth and increase the signal to noise ratio.

- p. 4: 'the ILS . . . retrieved simultaneously . . .': Since differences to the ideal ILS are hardly to distinguish with differences of the profile shape it is strongly recommended to retrieve the ILS from cell spectra. How does the resulting ILS looks like? Does it differ with respect to the ideal ILS and how much does it vary with time?

Technical corrections:

- p. 3, line 1: in June at Xianghe => at Xianghe in June

- p. 4, line 15: O3 retrieved profiles => retrieved O3 profiles

- p. 6, line 23: mainly the => mainly from the

- p. 7, line 4: larger the => larger as compared to the

- p. 16, line 2: a MCT => an MCT?

---

## Author Response (AR1)

Black: referee's comments red: authors' answers First of all, we want to thank the two referees for the detailed analysis of our paper. For the details, please look into the paper with keeping track of changes.

Referee #1

General comments: This manuscript presents a generally well written study on ozone retrievals from ground-based Fourier-transform infrared (FTIR) solar absorption spectra in the 3040 cm-1 spectral region. This is not a new approach, it was previously used (Rinsland et al., 1996; Goldman et al., 1999; Meier et al., 2005; Fu et al., 2007; Sung et al., 2007) for ozone retrievals, and as the study shows, this is not an optimal region for retrieving ozone. The 1000 cm-1 region proved to be more adequate for this purpose (Lindenmaier et al., 2010), and was adopted by the Network for the Detection of Atmospheric Composition Change for its harmonized FTIR ozone retrieval strategy (Vigouroux et al., 2015). However, for this particular site, Xianghe, where the spectral range is limited to the 1800 - 11000 cm-1 domain, it can give useful information about the seasonal ozone variations and its long-term trends. This retrieval approach can be extended to other FTIR sites recording spectra in this range. Therefore, I recommend this study for publication in AMT after minor revisions.

1) P3L25 – You mention "One specific optical bandpass filter. . ..." Can you please be more specific? Is that the standard narrow bandpass Filter 3 (2420-3080 cm-1) used by the NDACC-IRWG community? Is it wedged?

Yes. The filter used in Xianghe is the standard narrow bandpass Filter 3 (2420-3080 cm-1) used by the NDACC-IRWG community, and it is wedged. This information is added in the paper.

2) P4L2 – Explain what is epsilon. Done

3) P4L12-14 – What was the criteria for choosing these three particular windows for your retrievals? In the 2000 – 4000 cm-1 there are other windows that could be used for ozone retrievals, e.g. 2775 cm-1, 3023 cm-1. Also, you affirm that the first window has the strongest ozone absorption lines and the least interference with H2O. Why did you add the other two? Wouldn't have been enough to use only the first? You should explain in the text. Also, clarify if these windows were used simultaneously.

Thanks for the comments/suggestions. More texts are added in the revised version.

Comparing to 2775 cm-1, the  $O_3$  lines around 3040 cm-1 have stronger intensity. Comparing to 3023 cm-1, the  $O_3$  lines around 3040 cm-1 have less impact from other species, especially from  $H_2O$ . The retrieval windows used in this study are basically from Garcia's choise with some modification. We replace one of their windows with the window 1 in this study, which has less  $H_2O$  influence. The comparison between the retrieved  $O_3$  total column using Garcia's windows and our windows is added in the revised paper: it is found that the retrieved  $O_3$  total columns from Garica's windows and our choices are very close to each other. However, we have more successful retrievals when using the windows in this study compared to the one using Garcia's window.

The 3 retrieval windows are used simultaneously. Using three window together allows us to get a larger DOFS (2.4) compared to only using first window (1.5). According to Figure 1, the first window has the strongest ozone absorption lines and the least interference with  $H_2O$ , but

the latter two windows have more weak absorptions, which have more information in the stratosphere.

4) P4L19-20 – Have you tried fitting the minor interfering species to improve the residual? For example, for the 3039.9 – 3040.6 cm-1 window, what is the result if you fit also CH3Cl? Solar lines are not mentioned at all in the text, only in the caption of Figure 1. Among the weak species for this same window, beside solar lines you have HDO, NH3, and OH. Have you tried fitting these species? It would be great to add some text here and explain how you picked the interfering species for each window rather than just list them.

Thanks for the suggestions. Solar lines are now added in the text.

We have tested fitting the minor interfering species to improve the residual, for example adding  $CH_3Cl/HDO/NH_3$  and OH in the window 1, the largest improvement of the RMS is less than 0.001% and the change of the retrieved O3 total column is within 0.01%. Considering the relatively large systematic and random uncertainties of the O3 retrieved column of 13.7/1.4%, these weak species can be ignored.

5) P5 Figure 1 – Please enlarge the panels for each window (make them as those in Figure 2 for clarity. Also, the values on the x and y axes are too small, hard to read. Bring them at the size in Figure 2.

**Done**

6) P9L15-20 – This part is confusing. What is the accuracy and precision of the IAP ozonsondes? What does "higher ozone detecting performance" mean? I would give numbers here, the error for the IAP ozonsondes.

Thanks for the comments. Numbers are added in the revised version. The precision of the IAP ozonsondes is within  $\pm 5\%$  in the troposphere and within  $\pm 10\%$  in the stratosphere, respectively.

7) P10 Figure 4 – Please enlarge the numbers on the x and y axes Done

8) P11L6 – FTIR measurements are compared with TROPOMI OFFL at both sites, but for what window? Specify. Added

9) P16L8 – To me it looks like it is more 10 to 40 km rather than 5 and 40 km (text). In my opinion it is not correct then to use surface to 20 km. Use 10 to 20 km in the entire manuscript, I think it is more appropriate.

Done. We change '5 and 40 km' to '10 to 40 km'.

We prefer to keep the partial column between the surface and 20 km. As the DOFS for the partial column between surface and 20km is about 1.1 (see Fig. 2), including 0.25 from surface to 10 km and 0.85 from 10 to 20 km. To have >1.0 DOFS, it is better to use the partial column between surface and 20 km. We agree with the referee that the  $O_3$  retrieval (3040 cm-1) is mainly sensitive to 10-40 km, and we highlight in the paper that the lower partial column (surface-20 km) is mainly sensitive to the upper troposphere and lower stratosphere (UTLS), and less sensitive to the boundary layer.

There are some typos in the text: P2L15 – Change "the continue" to "to continue" P6L24 – Change "mainly the" to "mainly from the" P13 Figure 6 caption L5 – Change "and the back solid line" to "and the black solid line" Corrected

References:

Rinsland CP, Connor BJ, Jones NB, Boyd I, Matthews WA, Goldman, A, et al. Comparison of infrared and Dobson total ozone columns measured from Lauder, New Zealand. Geophys Res Lett 1996; 23:1025–8.

Goldman A, Paton-Walsh C, Bell W, Toon GC, Blavier JF, Sen, B, et al. Network for the Detection of Stratospheric Change Fourier trans- form infrared intercomparison at Table Mountain Facility, Novem- ber 1996. J Geophys Res 1999; 104:30481–503.

Meier A, Paton-Walsh C, Bell W, Blumenstock T, Hase F, Goldman, A, et al. Evidence of reduced measurement uncertainties from an FTIR instrument intercomparison at Kiruna, Sweden. JQSRT 2005; 96:75–84.

Fu D, Walker KA, Sung K, Boone CD, Soucy MA, Bernath PF. The portable atmospheric research interferometric spectrometer for the infrared, PARIS-IR. JQSRT 2007; 103:362–70.

Sung K, Skelton R, Walker KA, Boone CD, Fu D, Bernath P. N2O and O3 Arctic column amounts from PARIS-IR observations: retrievals, characterization and error analysis. JQSRT 2007; 107:385–406.

Lindenmaier R, Batchelor RL, Strong K, Fast H, Goutail F, Kolonjari F, et al. An evaluation of infrared microwindows for ozone retrievals using the Eureka Bruker 125HR Fourier transform spectrometer, JQSRT 2010; 111(4):569-585.

**Black: referee's comments red: authors' answers**

*First of all, we want to thank the two referees for the detailed analysis of our paper. For the details, please look into the paper with keeping track of changes.*

**Referee #2**

General comments: The authors present a study on ozone retrievals from infrared spectra recorded in Xinghe, China and on Reunion Island. Data from these sites are highly needed since these areas are poorly represented in the networks. This study uses the 3040 cm-1 spectral region and presents results of a one year time series and a characterisation of the 3040 cm-1 ozone product. Moreover, using spectra from Reunion Island data obtained from the 3040 cm-1 region are compared with those with the standard NDACC retrieval at 1000 cm-1. The comparison shows a good correlation, but a bias of 5.5 to 9.0 % and reduced degrees of freedom compared to the standard microwindow. Ozone retrievals in the 3040 cm-1 region are very useful since there are several FTIR spectrometers without an MCT detector around the globe.

For the 3040 cm-1 retrieval a modified version of the recipe of Garcia et al., 2014, was used. As a result, the key findings are very similar to those obtained by Garcia et al., 2014. However, since the recent study doesn't use exactly the same recipe, strictly speaking, it cannot be used as confirmation of the Garcia recipe and as an extension including more sites covering different conditions. To my impression, it is not clear whether it is a confirmation of the Garcia paper showing similar retrieval results or whether there is an improvement as compared to the Garcia paper. If the authors claim the latter this should be demonstrated or at least discussed in detail. To do so the authors might think in adding a Garcia type retrieval for comparison.

Therefore, I would recommend publishing this paper after major revisions although the paper is well written and fits well to the scope of AMT. Please also see specific comments below.

Specific comments:

- The statement in the abstract 'as the harmonized . . . uses the 1000 cm-1 spectral range, we designed an alternative O3 retrieval strategy . . .' is not correct since there is a published 'alternative' retrieval recipe for FTIR sites without MCT detector as published by Garcia et al., 2014.

Thanks for pointing out the inappropriate statement.

The sentence is reworded in the revised version "we apply the O3 retrieval in the 3040 cm-1 spectral range at Xianghe."

- The recipe from Garcia et al. 2014, has been modified. The modifications made and the rationale behind these modifications should be described in more detail. Moreover, a comparison with retrieval results using the full recipe from Garcia et al. would be very useful to see the effect of these modifications.

Thanks for the suggestion. The comparison between the FTIR  $O_3$  retrievals using the window in this study and García et al., 2014 window has been added in the Appendix A of the revised paper.

In general, the retrieved  $O_3$  total columns at Xianghe using the windows in this study and the window from García et al. 2014 are very close to each other. The mean and standard deviation of their relative difference are 0.8% and 1.2%, which are quite small compared to the retrieval uncertainty. However, we have more successful retrievals when using the windows in this study compared to their window choice, especially in summer with more H2O. The RMS of the residual using the windows in this study is about 0.20%, which is less compared to the one using Garcia's window of about 0.24% mainly due to several bad CH4 fittings. In addition, the mean of daily standard deviation of the retrieved total column for all days with more than 4 measurements using the García's window is 1.4%, which is slightly larger compared to 1.3% using the windows in this study. As the water vapor abundance is relatively high in summer at Xianghe, we suggest using the window of 3039.9-3040.6 cm-1 instead of the window of 3042.48-3043.72 cm-1.

- p. 4: 'a few badly fitted absorptions': Fig. 1 shows strong residuals at ozone line positions in particular in microwindow 1, not included in the Garcia paper. Does this additional window really improves the fit results although the line list needs improvement for this window? Thanks for the comments. By comparing FTIR  $O_3$  [3040 cm-1] retrievals with other datasets (FTIR  $O_3$  [1000 cm-1] retrievals, FTIR  $O_3$  [3040 cm-1] retrievals using García's window and TROPOMI measurements), it is found that the FTIR  $O_3$  [3040 cm-1] retrievals are generally in good agreement with other datasets apart from a systematic uncertainty. Adding the microwindow 1 does not harm the retrieval, although the  $O_3$  lines are not perfectly fitted. On the contrary, by adding the microwindow 1, the  $O_3$  retrieval has more information in the troposphere due to a stronger  $O_3$  line intensity compared to the lines in microwindows 2 and 3. The averaged DOF is 2.2 using only bands 2 and 3, and the DOF is 2.4 using 3 bands together at Xianghe.

- p. 3: 'One specific optical bandpass filter (2000 – 4000 cm-1)': This is not the standard NDACC type optical filter. The NDACC type filters provide a smaller bandwidth and increase the signal to noise ratio.

The filter used in Xianghe is the standard narrow bandpass Filter 3 (2420-3080 cm-1) used by the NDACC-IRWG community, and it is wedged. This information is added in the paper.

- p. 4: 'the ILS . . . retrieved simultaneously . . .': Since differences to the ideal ILS are hardly to distinguish with differences of the profile shape it is strongly recommended to retrieve the ILS from cell spectra. How does the resulting ILS looks like? Does it differ with respect to the ideal ILS and how much does it vary with time?

Thanks for the comments.

Simultaneous retrieving ILS allows us more freedom to fit the residual. We tune the sigma of the ILS parameter in sfit4.ctl to constrain the retrieved ILS and to make it close to the ILS results derived from the LINEFIT using the HBr cell measurements. Figure 1a shows the modulation efficiencies (ME) retrieved by the LINEFIT14.5 code from 4 HBr cell measurements at Xianghe. Figure 2 shows an example of the a priori and retrieved ME, as well as the time series of the retrieved ME at the maximum optical path difference (MOPD = 175 cm). The a priori ME is the ideal status, and the retrieved ME at the MOPD are 0.88 and 0.04, respectively, and the retrieved ME is relatively stable with time.

The LINEFIT retrieval also suffers from the uncertainties of the cell pressure, temperature and gas abundance, and it is not easy to estimate these uncertainties. Therefore, we prefer to

retrieve the ILS but with a reasonable sigma to constraint the retrieved ILS parameters and to make them close to the cell measurements instead of using the LINEFIT outputs directly.

Figure 1a. The modulation efficiencies retrieved by the LINEFIT14.5 code from HBr cell measurements at Xianghe on 7 June 2018, 9 October 2018, 18 July 2019 and 20 December 2019.

Figure 2a. Left panel: a typical example of the a priori and retrieved modulation efficiencies (ME) along with the optical path difference (OPD) at Xianghe. Right panel: the time series of the retrieved ME at the Maximum OPD (175 cm).

Technical corrections:

- p. 3, line 1: in June at Xianghe => at Xianghe in June
- p. 4, line 15: O3 retrieved profiles => retrieved O3 profiles
- p. 6, line 23: mainly the => mainly from the
- -p. 7, line 4: larger the => larger as compared to the
- p. 16, line 2: a MCT => an MCT?

Corrected

[revised manuscript text omitted]

$$\boldsymbol{J}(\boldsymbol{x}) = [\boldsymbol{y} - \boldsymbol{F}(\boldsymbol{x})]^T \mathbf{S}_{\boldsymbol{\epsilon}}^{-1} [\boldsymbol{y} - \boldsymbol{F}(\boldsymbol{x})] + [\boldsymbol{x} - \boldsymbol{x}_{\boldsymbol{a}}]^T \mathbf{S}_{\boldsymbol{a}}^{-1} [\boldsymbol{x} - \boldsymbol{x}_{\boldsymbol{a}}],$$
(2)

5 where y and F(x) are the observed and fitted spectra, respectively, Sε is the measurement covariance matrix and Sa is the a priori covariance matrix. J(x) is the combination of the measurement information and the a priori information, with their weightings determined by Sε and Sa. Sε is derived from the SNR of the spectra, with its diagonal values set to 1/SNR2 and off-diagonal values to 0. Sa is derived from the covariance matrix of the Whole Atmosphere Community Climate Model (WACCM) v6 O3 monthly means between 1980 and 2020. The square root of the diagonal elements of Sa are about 3% near the surface, 2% in the troposphere, 2.5% in the stratosphere and 1% above the stratosphere.

Table 1 lists the parameters adopted in the retrieval strategy for the FTIR  $O_3$  measurements at Xianghe in this study. We selected three retrieval windows (3039.9 - 3040.6 cm-1, 3041.5 - 3042.25 cm-1 and 3044.7 - 3045.54 cm-1) in this study, where the latter two windows are taken from the study of García et al. (2014); the first window has the strongest  $O_3$  absorption lines and the least interference with H2O. Comparing to the retrieval windows used in García et al. (2014), the FTIR  $O_3$

- 15 retrieved total columns from the three windows in this study are similar but slightly less affected by H2O abundances , and the O3 retrieved profiles are less oscillating by comparison with ozonesonde profiles at Xianghe and Maïdo(see Appendix A). For the spectroscopic data, we use the atmospheric line list ATM2019 (https://mark4sun.jpl.nasa.gov/pseudo.html; last access: 26 March 2019). Figure 1 shows an example of the absorption lines and residuals in the three retrieval windows at Xianghe. The root mean square (RMS) of the residual is about 0.2%. It contains a few badly fitted absorptions at O3 line positions in
- 20 these 3 windows, caused by uncertainties in the spectroscopy. Further investigations are needed to improve the spectroscopic parameters in this spectral range, but that is beyond the scope of this study. To reduce the influence from the interfering species, CH4, HCl, H218O, H217O, H2O, HDO and CO2 columns are retrieved simultaneously with the O3 profile. The specific interfering species are selected for each window (see Figure 1), because they have relatively larger absorptions compared to other weak species, e.g. CH3Cl, NH3 and OH. In addition, the solar intensity and wavenumber shift are retrieved simultaneously. Note that
- 25 the H2O isotopes (H $_2^{18}$ O, H $_2^{17}$ O and HDO) are treated as individual species in the SFIT4 algorithm. The instrument line shape (ILS) is part of the state vector and retrieved simultaneously along with the O3 profile, with an ideal ILS being applied as the a priori input.

The temperature, pressure and  $H_2O$  profiles are from the National Centers for Environmental Prediction (NCEP) 6-hourly re-analysis data. For the a priori profiles of  $O_3$  and other interfering species, we use the mean of the WACCM model data

30 between 1980 and 2020. Since the broadening effect of absorption lines is related to the pressure and temperature, we can obtain limited vertical information of  $O_3$  by fitting the spectra. Figure 2 shows an example of the typical averaging kernel of the FTIR  $O_3$  retrieval at Xianghe. The retrieved  $O_3$  profile is mainly sensitive to the vertical range between 5-10 and 40 km. The degree of freedom for signal (DOFS) is  $2.4 \pm 0.3$  (1 $\sigma$ ), indicating that there are two individual pieces of information:

**Figure 1.** Example of spectral fits in the three microwindows for  $O_3$  retrievals at Xianghe. Lower panels: the normalized transmittance from each atmospheric species and solar lines. Upper panels: the difference between the observed and fitted spectra (Obs-Fit).